# Numerical Simulation of Oil Spills in the Lower Amazonas River

Sarana Castro Demoner [1,*], Marcelo Rassy Teixeira [1], Carlos Henrique Medeiros de Abreu [2] and Alan Cavalcanti da Cunha [3,4]

1 Graduate Program in Dam Engineering and Environmental Management (PEBGA), Federal University of Pará (UFPA), Tucuruí 68464-000, Brazil
2 Environmental Engineering School (CEAM), State University of Amapá (UEAP), Macapá 68900-070, Brazil
3 Graduate Program in Tropical Biodiversity (PPGBIO), Graduate Program in Environmental Science (PPGCA), Environmental Science Course (CA), PPG-Bionorte, Federal University of Amapá (UNIFAP), Macapá 68903-419, Brazil
4 Graduate Program in the Legal Amazon Biodiversity and Biotechnology Network, State University of Amapá (UEAP), Macapá 68900-070, Brazil
* Correspondence: saranatuc@unifesspa.edu.br

**Abstract:** In 2013, a slope slide took place at the Santana-AP channel that links to the Lower Amazon River's North Channel, resulting in the sudden collapse of a substantial section of the Port of Santana and its infrastructure. This area houses liquid bulk terminals and pipelines with high pollution potential. The objective of the research is to evaluate the potential environmental impacts of an eventual oil spill in the very short term using a numerical hydrodynamic simulation model coupled with that of pollutant dispersion. The SisBaHiA® software, experimentally calibrated using acoustic methods (ADCP), was used to generate hypothetical scenarios in these areas with a substantial risk of landslides. Two hydrological scenarios stand out in the simulations: (a) November S-1 (dry) and (b) May S-2 (rainy). In S-1, the plume dispersion was higher during flood tides S-1a and S-1b, reaching 4 h urban slope areas, river mouths, tributaries (Matapi and Vila Nova), and environmental protection areas. At S-2, the plume spread was greater during the ebb tides S-2c and S-2d, affecting Macapá's water supply system 12 h after the accident. The scenarios suggest the existence of high risks associated with the study hypotheses. The dispersion of the plume is directly proportional to the flow, indicating that local hydrodynamics is probably the most relevant dispersive factor. We conclude that the mitigation time for more severe effects is critical in the first 4 h because the coastal geographic feature tends to keep the plume in the Santana channel.

**Keywords:** environmental impact prevention; port monitoring; SisBaHiA

## 1. Introduction

The negative environmental impacts caused by possible heavy oil spills in ports and waterways have been frequent, although avoidable, and their main consequence is the lasting pollution of water and biota [1,2]. However, the bureaucracy and slowness of the governmental systems responsible for activating control plans, when they exist, may be the main impediments to mitigation actions, especially in the short term, since they require reliable simulation models to represent the displacement of plumes of oil or oil byproducts [1–3].

Environmental risk assessment is essential in areas with fuel reserves [4]. For example, there was an unforeseen and sudden slope rupture on the left bank of the Santana Canal in Amapá, specifically in the area where the Port of Santana's amenities are located. The accident provided no previous signs of ruptures, cracks in the terrain, or even during the movement of the facilities, but caused a serious, unprecedented accident with significant material and economic losses, including human lives and numerous environmental consequences [5,6]. The identification of the geotechnical characteristics of the site, however,

raised an alert of imminent potential risk linked to the existence of a pipeline network installed next to the scar formed by this accident [6].

Additionally, according to the tests of forensics carried out at the site after the accident [6], a sensitive soil known as "quick clays" was identified, which have a rheological behavior similar to that of fluid when stressed [6–8]. Thus, scientific evidence revealed unusual behavior in the local soil of the Porto de Santana accident [6]. Hence, the need for specific studies that provide answers to the following questions: (a) What are the proportions and scope of the potential environmental impacts in the event of an accident similar to the one observed in the landslide but involving local pipelines, installations, and fuel reserves? (b) What areas and critical moments would be most affected by the oil plume along the specific coastal zone in the region of the mouth of the Amazon River? (c) What is the tactical response time that those responsible would have to take to initiate mediation, containment, and mitigation actions for the impacts caused by an eventual accident of this proportion?

In recognized poorly monitored aquatic environments, both hydrologically and in terms of water quality, computer modeling, and simulation have been used to combine different variables and estimate oil spill scenarios with high accuracy, presenting sufficiently realistic results for such purposes [9,10]. Therefore, it is an alternative but highly accurate tool to guide actions concerning the containment and mitigation of potential negative effects of the oil plume as a preventive and tactical planning action in accident-risk areas [9,10].

This research aims to understand and simulate very short-term numerical scenarios to predict the spatial–temporal displacement of oil plumes caused by eventual spills in the Santana Channel. These spills are hypothetically caused by a slope rupture in the pipeline's risk area. The objective is to gain insight into the oil plume's behavior at the Amazon River's mouth. To achieve this, hypothetical scenarios were modeled and simulated using SisBAHIA 11 software. This software allows for modeling different representative hydrodynamic and environmental scenarios contained in the local geography.

In addition, the research had specific objectives: (a) to identify critical contamination zones and sensitive areas potentially impacted, as suggested by the Atlas of Environmental Sensitivity to Oil of the Foz do Amazonas Maritime Basin. This includes assessing the environmental impact along the coast [9,11]. (b) To propose prevention strategies and specific, very short-term mitigating actions against accidents in the port area of Santana-AP. The main justification is that this area represents a higher potential risk of accidents because it is close to the scar generated by the 2013 accident and is located near the two most populous municipalities of the state of Amapá (Macapá and Santana).

## 2. Materials and Methods

The present study followed three main steps: (a) a data survey based on the environmental forensics performed in 2019 due to the environmental accident in the port of Santana [6] and the research coming from that forensics [9,12,13]; (b) a computational simulation of the observed hydrodynamics in May (rainy period) and November (dry period) (flow, tidal elevation, and current velocity); and (c) a hydrodynamic and Lagrangian transport model simulation for the pre-established scenarios involving potential claims in the environmental risk area. The experimental survey campaigns in 2019 provided detailed data on the discharge of water into the channel within a complete tidal cycle (12.5 h), in addition to bathymetry and velocity characteristics. Building a solid base of experimental information is necessary for model calibration and scenario simulation [9,12,13].

### 2.1. Characterization of the Study Site

The research covers the North Channel of the Amazon River, precisely the Santana Channel, assuming as a plume release point the end of the pipeline that intersects the channel. This stretch of land exhibits a flat hydrogeomorphology and is heavily influenced by both fluvial processes (such as the North Channel of the Amazon River) and coastal processes (such as semidiurnal mesotides with an approximate amplitude of 2.8 m). These

dynamics often lead to the escalation of erosion and sedimentation [12,14]. The Amazon Coastal Zone has areas of high vulnerability associated with the sensitivity of the dominant estuarine and oceanic ecosystems (mangroves, riverbanks, floodplains, marshes, and vegetated islands) [13,15].

*2.2. Modeling Conditions*

Sea and land boundary conditions, bathymetry data, and equivalent bottom roughness were used to make the modeling domains. The hydrodynamic model was forced through tide and wind inputs. The model was calibrated using current data obtained from an Acoustic Doppler Current Profiler (ADCP) provided by the Laboratory of Hydraulics and Environmental Sanitation, part of the Civil Engineering Course of the Federal University of Amapá (UNIFAP). The ADCP was used to quantify and determine the liquid discharges and current profiles throughout a complete tidal cycle, based on previous studies by Abreu et al., 2020 [12] and Cunha et al., 2021 [9].

The spill volume used in the modeling is based on the recorded flow rate for the pipeline, as per ANP operating authorization No. 468, OF 15.10.2012, of 800 m$^3$/h when in transport operation to onshore reservoirs. The scenarios were distributed over brief time intervals after the spill of 1, 4, 8, and 12 h. Each time interval was analyzed according to climatic characteristics: one scenario in a predominantly dry Amazonian hydrological period (November) and the second in a predominantly wet period (May). Each seasonal period totaled four scenarios. Considering the seasonal hydrological variation, the amplitude of variation, the spatiotemporal scope, and the accuracy of the hypothetical behavior of the plume under different environmental conditions are inferred.

The wind speed intervals were obtained from an 8-year time series (2010–2018) made available by the Macapá weather station (MACAPA-AP-OMM: 82098), located near the Fazendinha APA. The meteorological data were collected in the Meteorological Database for Teaching and Research (BDMEP) of the National Institute of Meteorology of Brazil (Instituto Nacional de Meteorologia—INMET) and present precipitation records between ≈400 mm in the rainy season and ≈60 mm in the dry season, while the wind intensity remained between the ranges: $[0 \leq V_{(rainy\ wind)} \leq 2\ m\ s^{-1}]$ e $[1 \leq V_{(dry\ wind)} \leq 3\ m\ s^{-1}]$ at the respective stations.

The hydrodynamic model and transport scenarios were generated by the software SisBaHiA®—Base System of Environmental Hydrodynamics, which uses a 3D or 2DH hydrodynamic circulation model optimized for natural water bodies, as well as Eulerian and Lagrangian models (pollutant dispersion) for transport phenomena [10]. The 3D model (Equations (1)–(4)) was used to simulate the hydrodynamic behavior of the current, while the Lagrangian model was used to simulate its dispersive process and plume behavior (coupled to the hydrodynamic one). The position of any particle at the next instant ($P^{n+1}$) is determined by a second-order expansion in the Taylor series according to the previously recorded position ($P^n$). This transport is represented by several immaterial particles transported by currents defined by the hydrodynamic model [9,10,12,13].

$$\frac{\partial u}{\partial t} + u\frac{\partial u}{\partial x} + v\frac{\partial u}{\partial y} + w\frac{\partial u}{\partial z} = -g\frac{\partial \zeta}{\partial x} - \frac{1}{\rho 0}\,g\int_z^\zeta \frac{\partial \rho}{\partial x}\,dz + \frac{1}{\rho 0}\left(\frac{\partial \tau_{xx}}{\partial x} + \frac{\partial \tau_{xy}}{\partial y} + \frac{\partial \tau_{xz}}{\partial z}\right) + 2\Phi \sin\theta v \tag{1}$$

$$\frac{\partial v}{\partial t} + u\frac{\partial v}{\partial x} + v\frac{\partial v}{\partial y} + w\frac{\partial v}{\partial z} = -g\frac{\partial \zeta}{\partial y} - \frac{1}{\rho 0}\,g\int_z^\zeta \frac{\partial \rho}{\partial y}\,dz + \frac{1}{\rho 0}\left(\frac{\partial \tau_{yx}}{\partial x} + \frac{\partial \tau_{yy}}{\partial y} + \frac{\partial \tau_{yz}}{\partial z}\right) - 2\Phi \sin\theta u \tag{2}$$

$$\frac{\partial \zeta}{\partial t} + \frac{\partial}{\partial x}\int_{-h}^\zeta u\,dz + \frac{\partial}{\partial y}\int_{-h}^\zeta v\,dz = q_p - q_E + q_{Ia} - q_{Ie} \tag{3}$$

$$\frac{\partial u}{\partial x} + \frac{\partial v}{\partial y} + \frac{\partial w}{\partial z} = 0 \tag{4}$$

$$P^{n+1} = P^n + \Delta t\frac{dP^n}{dt} + \frac{\Delta t^2}{2!}\frac{d^2 P^n}{dt^2} + O^3 \tag{5}$$

where $u$, $v$, and $w$are, respectively, are the components of the velocity vector in the $x$, $y$, and $z$ directions in (m/s). The vertical direction $z$ is positive upwards, and its origin can be conveniently defined at the mean water surface level. $P$ is the pressure, $\rho$ is the local specific mass of the fluid (kg m$^{-3}$), and $\rho 0$ is a constant reference specific mass (kg m$^{-3}$). $\Phi$ is the Earth's rotational angular speed in the local coordinate system (rad s$^{-1}$), and the terms with $\Phi$ are the Coriolis forces, where $\theta$ is the Latitude angle. $\zeta$ represents the free water surface elevation (m), and the components ($q_p$, $q_E$, $q_{Ia}$, $q_{Ie}$) represent the balance of precipitation, evaporation, and inflow and outflow fluxes, respectively, per unit area [9,10,12,13].

The domain mesh of the computational model consists of 2238 elements, generating 10,627 nodes that are distributed over an area of approximately $3 \times 10^7$ km$^2$ [9,12,13]. For the tide generation at the domain boundary, the study utilized the most relevant harmonic constants for the study site. These constants include M2, S2, N2, K1, O1, and M4. Among these constants, special emphasis was given to the M2 constant, which has the greatest influence on the Brazilian coast. The M2 constant is responsible for representing about 70% of the variation in the physical behavior of the tides in the region [16,17].

The behavior of the S500 oil (decay) required to feed the Lagrangian model of SisBaHIA was obtained using the ADIOS2 software, adopting a volume of 800 m$^3$ of persistent hydrocarbon with the following characteristics: $^\circ$API = 35.2, specific gravity = 0.840 kg m$^{-3}$ at 22 $^\circ$C, viscosity = 40 cSt at 22 $^\circ$C, and pour point = 38 $^\circ$C (ADIOS2, 2021). The UTM coordinates of the contaminant source release are the meeting of the pipeline at the edge of the Santana channel: X = 479,441.15 m E and Y = 9,993,724.29 m S. The continuous source release time was estimated at a maximum of 1 h. Therefore, the simulated oil plume in the present study was S500, the most common oil transported and stored in the Amazon [9].

The specialized literature describes a series of analyses that can be applied in calibration processes [12,18,19]. The calibration of the model used in the study used proximity mechanisms between simulated results and field measurements, considering the best similarity within acceptable ranges and adjusting the parameters that required calibration [12,19]. The objective of the model was to probabilistically simulate a random day of the month for the accidental oil spill. This hypothesis was based on the fact that an accident could occur at any time in this seasonal interval since the geotechnical characteristics represented by the sensitive clay were identified at the site. Therefore, it has not previously manifested physical signs of soil rupture that would serve as a warning [6–8].

The methodology employed, including the use of SiBahia and other auxiliary software, can be applied to any dynamic water body, provided that each stage of data collection and model calibration is carried out diligently. It is essential to gather accurate data specific to the water body and ensure that the model is appropriately calibrated. This allows the model to faithfully represent the hydrodynamic behavior of the water resource under investigation [10].

## 3. Results and Discussion

Two scenarios were simulated for each predominant hydrological period in the Amazon region. Namely: (1) dry period, the month of November is highlighted (scenario S-1); and (2) wet period, the month of May is highlighted (scenario S-2). Both periods were identified by the following characteristics of a normal local semidiurnal tidal cycle ($\approx$12:40 h): (a) beginning of the flood tide; (b) end of the flood tide; (c) beginning of the ebb tide; and (d) end of the ebb tide. Therefore, it comprises hypothetical plume behavior within a semidiurnal tidal cycle, generating up to four scenarios in each seasonal period studied.

The hypothetical accident is established at the beginning of the pipeline structure, installed in the Port of Santana (latitude: $-3.810178^\circ$, longitude: $-49.560781^\circ$). The presence of islands and two large tributaries (the Matapi River and the Vila Nova River) were considered influential initial and boundary conditions in the present simulation [9,13,20,21].

Figures 1–4 show the results of the modeled and simulated S-1 scenarios for November (dry) at 1 h, 4 h, 8 h, and 12 h, respectively, for the best possible visualization of the dispersive process. This temporal configuration of dispersion was considered the most con-

venient to evaluate the plume movement within a very short-term tidal cycle [9]. This very short timeframe served to establish crisis management parameters and potential mitigation actions promptly [22], immediately after a probable landslide that would potentially affect the facility to the point of leaking stored oil [6].

It is important to highlight the first hours after the accident [22]. During this period, the hydrodynamics and surface currents, which are influenced by the wind [20,23], give rise to a plume recirculation process around the discharge point. This phenomenon leads to increased contact between the banks and the contaminating particles, thereby amplifying the environmental impacts [9,22]. Additionally, since the dispersive process of pollutants is directly influenced by the hydrodynamics of these estuarine environments [9,21,24] and the effect of the winds [20,23], this interference also acts on evaporation rates and particle fragmentation, as emulsified surface oil would be more dispersed by the wind [25].

The initial hydrodynamic characteristics of the Santana Canal launch were the end of the ebb tide (left to right on the map), with a flow of 4125.75 m$^3$/s. Soon after this phase, the tide initiated the flood phase (reversal of flow) (right to left on the map). Accurately determining the specific stage of the tide during the hypothetical oil spill is essential for effective crisis management and the minimization of potential consequences. It allows for skilled and early recognition of the probable path of this contaminant in the very short term.

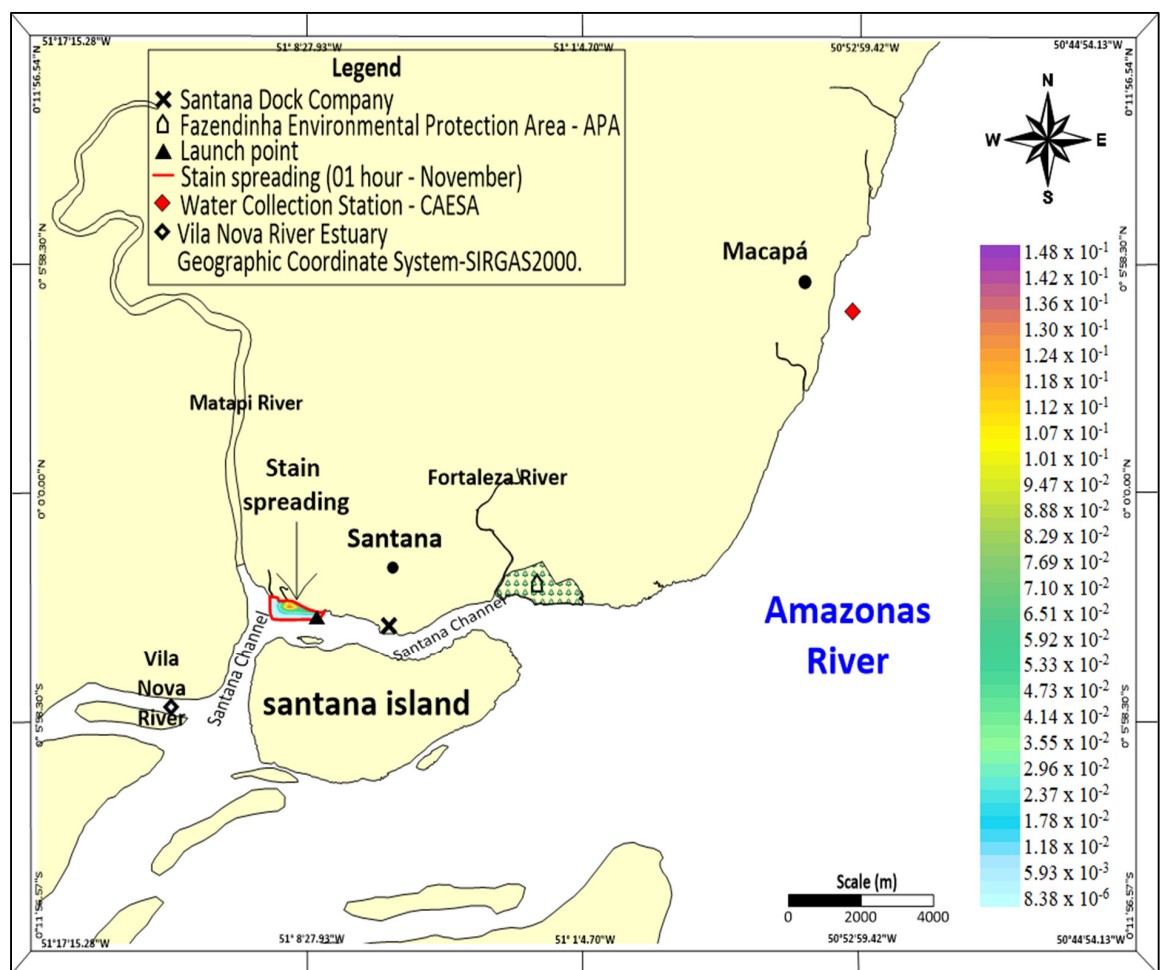

**Figure 1.** S-1a—1 h post-oil spill (November dry spell). Source: Author, 2022.

One hour after the accident in scenario S-1a (Figure 1), with the Santana Channel flow of 15,197.17 m$^3$/s, the slick occupied an area of approximately 0.53 km$^2$. In this configuration, the river current and tide were almost canceling each other out, directing the plume slightly to the west of the release point. It is also due to the absence of currents

or the effect of intense winds. The absence of interactions between these two factors may be due to the momentary absence of these forcings in this initial interval, favoring radial spreading (apparently more diffusive).

At time t = 4 h, referring to scenario S-1b (Figure 2), the flood tide flow intensifies (westward) when the plume is directed upstream of the dumping point, reaching an area of ≈11.27 km$^2$ and a Santana Channel flow of 10,194.36 m$^3$/s. The impacted areas up to this point were mainly on the west bank, with emphasis on the mouth of the Vila Nova River, a tributary of the Amazon River, where the highest concentrations of oil appeared.

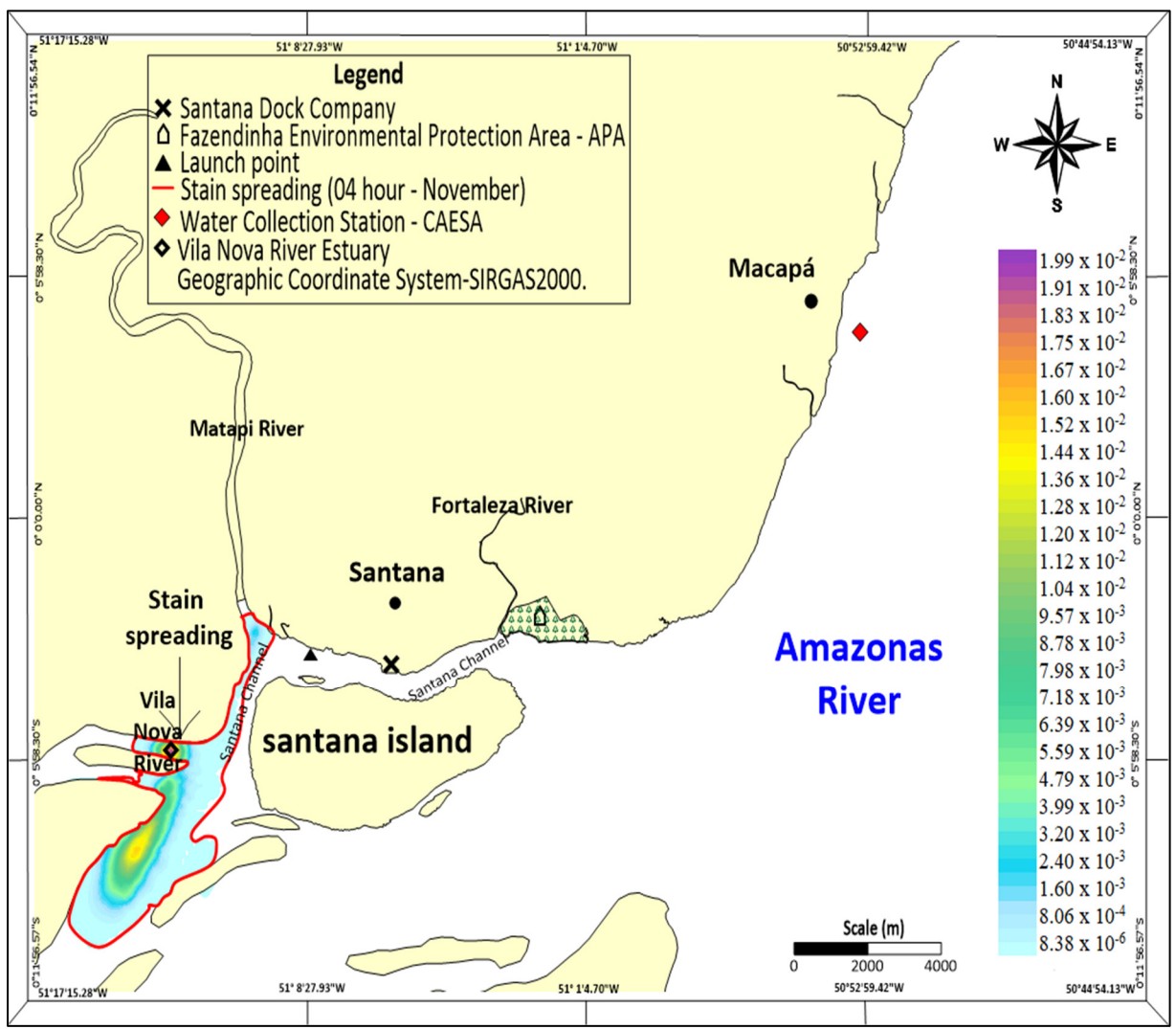

**Figure 2.** S-1b—4 h post-oil spill (November dry spell). Source: Author, 2022.

Hydrodynamic conditions, meteorological conditions, water residence time, and oil properties are characteristics that establish an intrinsic dynamism in hydrodynamic behavior and water renewal rates, influencing the dynamics of oil dispersion in the aquatic environment [9,12,13]. At this contact, the oil could already change the density and porosity of the soil, affecting the absorption of contaminants along the soil profiles reached [26–28]. Oil pollution in estuarine soils and wetlands represents a high ecological risk [9,11,13].

Therefore, the environmental response due to contamination implies a decrease in the bacteriological diversity of soils [29]. In aquatic ecosystems, it can interfere with the stable isotopic composition of dissolved O$_2$, which is regulated by air–water gas exchange processes, respiration, and photosynthesis [30], and microbial diversity is essential for biogeochemical cycles [29]. As a result, its immediate loss would imply the imminent

deterioration of the estuarine potential to activate essential ecological processes such as primary production [29,31,32].

In the 8-h analysis, scenario S-1c (Figure 3), the current starts to move towards the mouth of the Amazon but still concentrates in the port area in the Santana channel. The reached area was approximately 10.04 km$^2$, and the flow was 18,571.13 m$^3$/s. The ebb tide flow is starting its process due to the tidal change. However, because it is still at the beginning of the ebb tide, the flood tide (previous interval) and ebb tide (just started interval) flows still oppose each other. This balance of forces [33] partially retains the contaminant in the stretch between the mainland and Ilha de Santana. Therefore, the plume tends to remain partially trapped in the Santana Channel and neighboring islands, affecting both banks of the channel and the mouth of the Vila Nova River.

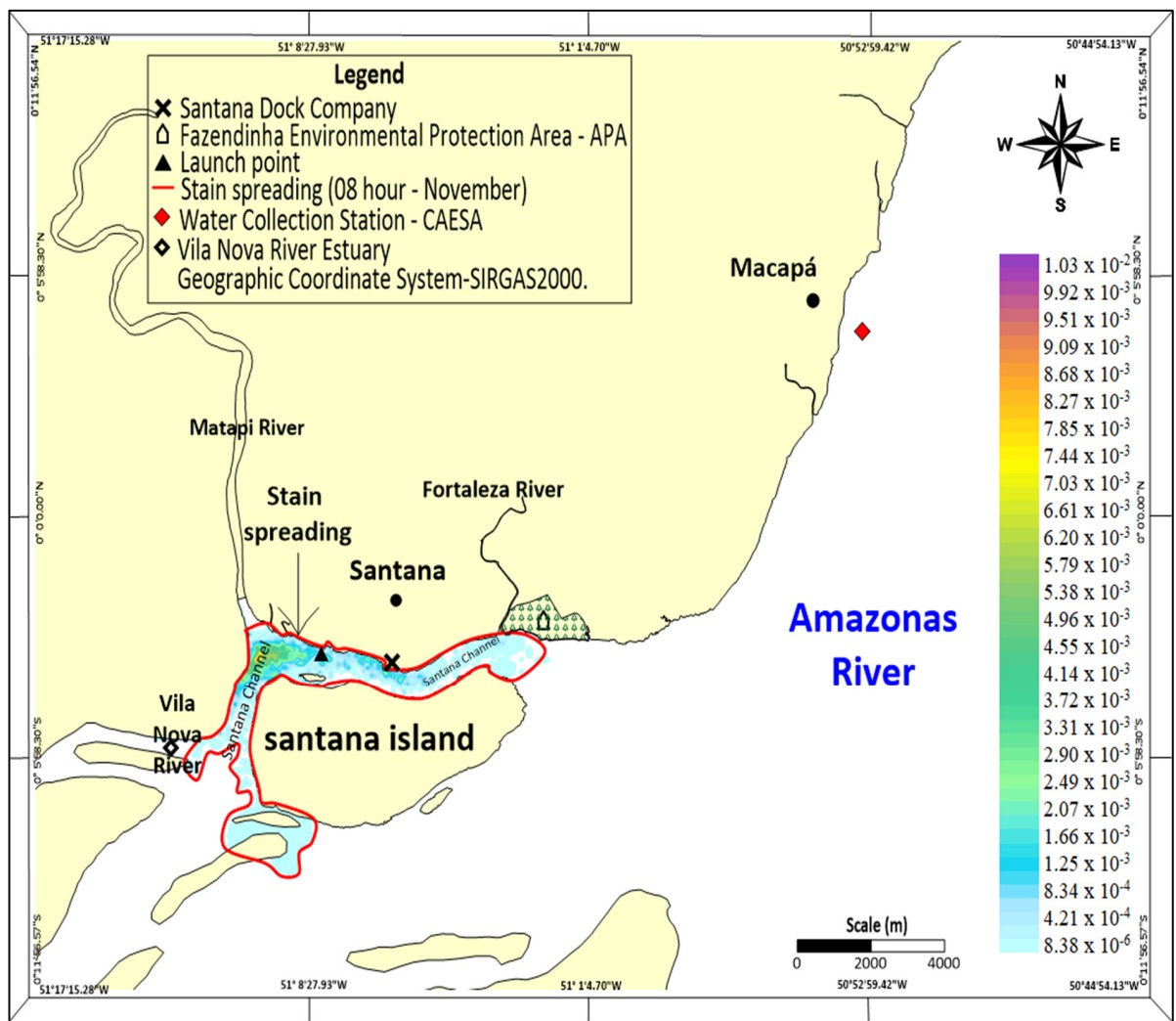

**Figure 3.** S-1c—8 h after the oil spill (dry season in November), Fonte: Author, 2022.

Under these circumstances, the plume would impact the entire local port area, the mouth of the Igarapé da Fortaleza, and urban areas of the cities of Macapá and Santana, including the Environmental Protection Area of "Fazendinha," whose border area coincides with the affected channel margin. In addition to the alterations in microbial processes that have been identified as having significant environmental impacts [29], it is important to consider that other external environmental factors, such as the hydrological regime, can also influence the degradation potential of petroleum products. For example, occasional flooding could promote or intensify anaerobic conditions at the sediment level and further hinder the biodegradation of hydrocarbons [34].

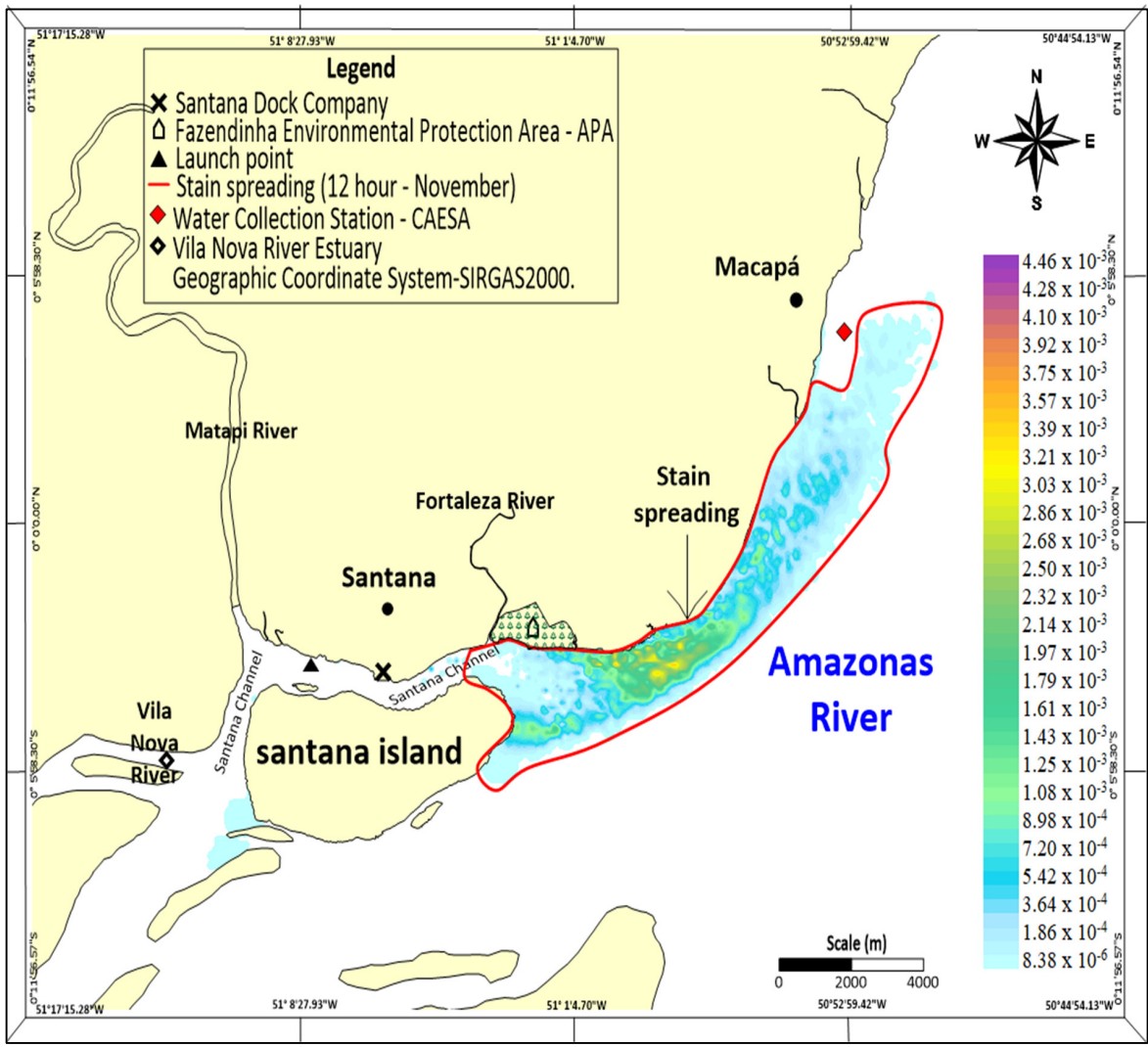

**Figure 4.** S-1d—12 h after the oil spill (dry season in November). Source: Author, 2022.

This scenario coincides with the sensitivity index of the study area [9,11,13] since estuarine environments have a wide hydrological variation [30], which characterizes their potential for natural self-depuration. In this sense, the contaminant can reach both the soil and the exposed roots of mangroves, including leaves and shoots [35]. Thus, the presence of oil in the environment causes severe adverse physical (photosynthesis), biophysical (cell membranes), and biochemical (enzyme inhibition) effects [29,34,35].

In the 12-h analysis, scenario S-1d (Figure 4), the ebb tide is still predominant with a flow of 11,362.22 m$^3$/s, intensifying the spreading of the particles to an area of 32.08 km$^2$. At this stage, the plume would reach the vicinity of the water catchment station of the water and sewage company of Amapá (CAESA). This probably changes the spring's water quality parameters and forces a greater quantitative and qualitative control over its potable conditions [13], or even a complete shutdown at a critical moment for safety and public health reasons.

This would trigger emergency action and a necessary mediation action for the oil spill [4,36]. For example, surfactants in dispersants contribute to the contamination of surface water and impair the water treatment process [35]. This provides operational and contingency upheaval beyond simple remediation due to potential initial oil contamination.

The affected shorelines in Scenario S-1d would be areas of multiple water resource use by the local community [11,13]. The Macapá water catchment system, the main one in Amapá State, is of importance for basic sanitation, economic subsistence, fishing,

recreation, and environmental protection [5,9,11,13]. The impacts are important because of the probable and evident contact with petroleum derivatives, which would be extremely harmful to human health because they are easily absorbed through the skin [37], ingested, or inhaled [37,38]. Furthermore, the resulting health problems would range from dermatitis to eye irritation, throat irritation, and even psychological problems [37]. However, such effects depend on the characteristics of the contaminant, its concentration, and the time of contact or exposure [38].

In the assessment interval between ≈4 h and ≈12 h, corresponding to Figures 2 and 4, respectively, the plume traveled approximately 26 km. The longitudinal distance of the plume varied from the mouth of the Vila Nova River to a significant stretch of the Macapá waterfront, almost reaching the Water Catchment Station. These results are similar to those presented by Cunha et al. (2021) [9], where the plume in this same interval traveled ≈25 km. The variations observed between the studies are primarily attributed to the differences in release points. However, the plume's overall behavior was similar and consistent between the two studies.

In scenario S-2, which was simulated for the Amazon rainy period (May), Figures 5–8 show the results for this period. The simulation took into account similar hydrodynamic characteristics of the tide as in November to enable a comparative analysis of plume spreading. However, during the actual spill, the flow rate of the Santana Channel was 6757.24 m$^3$/s, indicating the end of the ebb tide, similar to the S-1 scenario.

Despite the initial outflow from the Santana Channel being slightly higher than the November scenario, the plume showed little spreading in the first hour analyzed since the wind force had almost no major influence on contaminant entrainment in May [9]. Therefore, the dispersion in this scenario was mainly conditioned by the hydrodynamic characteristics, i.e., with negligible meteorological effects for this simulation [9,11].

In the analysis of 1 h S-2a (Figure 5), the plume reached an area of 0.34 km$^2$, and the Santana Channel flow was 12,103.63 m$^3$/s. This shows a sharp decrease compared to the S-1a scenario, even with equivalent intervals and tidal phases in both evaluations. Thus, the plume shape remained geometrically similar to the November period, only differing in proportion, which was corroborated by the low wind speeds recorded in May [9].

The slick in S-2a spread 36% less than in S-1a. This is due to the recorded flow in S-2a also being 21% less than in S-1a, contributing to the narrative of the influence of the hydrological regime on pollutant transport [8,10,12].

In the analysis corresponding to 4 h after the spill, representing the S-2b scenario (Figure 6), the plume followed the hydrodynamic behavior trend, following the direction of the most intense tidal currents. However, with a Santana Channel flow of 7716.667 m$^3$/s, the area reached by the spreading of the contaminant in this stage was only 2.54 km$^2$, reaching discretely the mouth of the Vila Nova River estuary. The oil plume's shyer behavior is a result of decreased wind influence and heightened susceptibility to surface water currents [9,23], making it more dependent on hydrodynamic factors [9].

Scenario S-2b also stands out for the difference in flow in the Santana Canal compared to the same moment in time in scenario S-1b, registering a 24.3% lower flow. Thus, their dispersion was 77% smaller when compared to each other. However, here we see an even more pronounced difference in the plumes for the same time interval after the hypothetical spill since the May plume already came from a scenario with less dispersion due to lower flow and less wind influence. This accentuated the difference between scenarios in the intervals analyzed later, generating a lower spread in the rainy season. It can be seen that the condition of hydrological saturation (river flooding) contributes to this lower variation of flow related to the flood tide, so that the forces resulting from the natural flow of the estuary would counterbalance each other [39,40]. Furthermore, the impact of the tributaries Matapi and Vila Nova's outflows cannot be overlooked as they peak during the ebb, effectively reducing tidal influence.

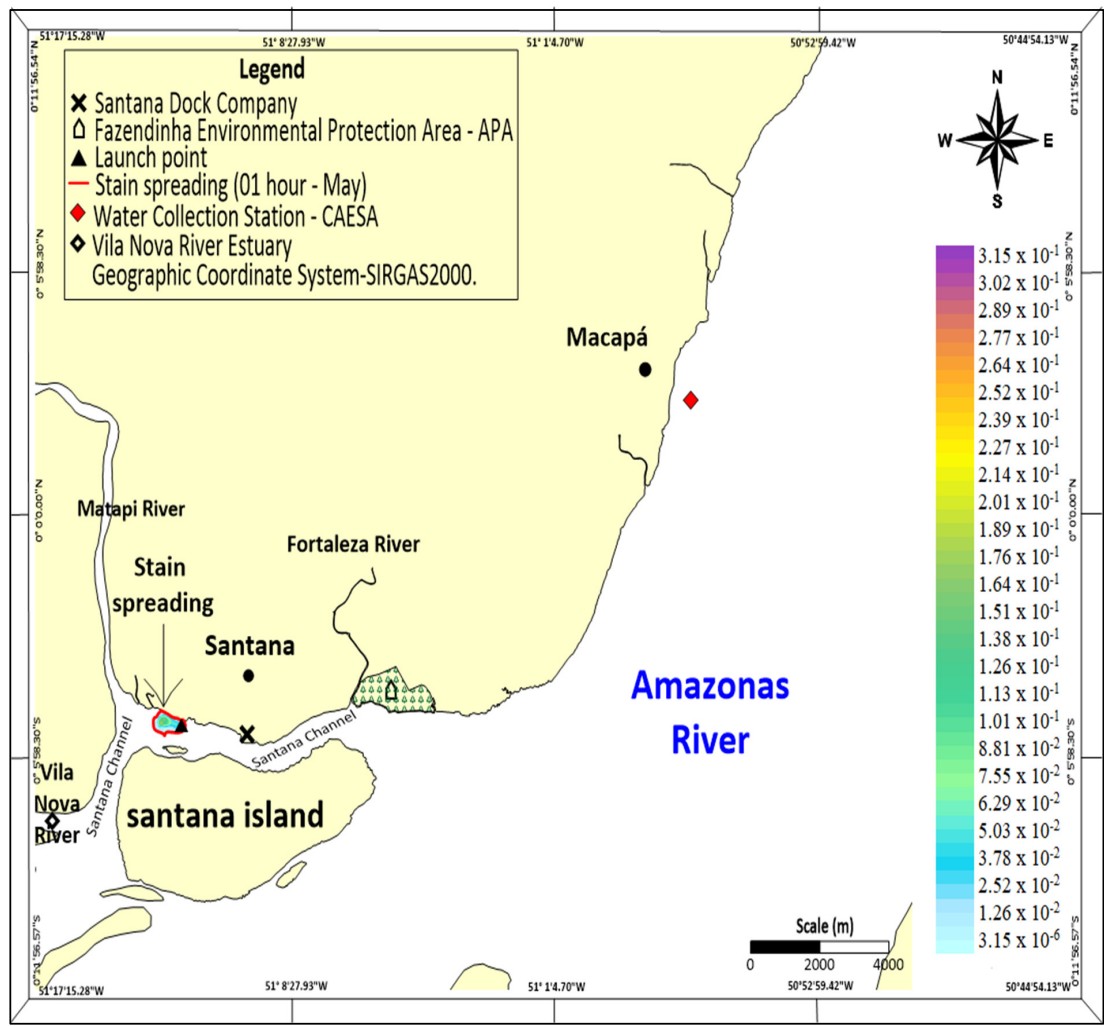

**Figure 5.** S-2a—1 h after the oil spill (rainy season in May). Source: Author, 2022.

It is important to emphasize that the influence of these two tributaries on the Santana Channel (Matapi and Vila Nova) is poorly known. While the Matapi River has been extensively researched, with a good understanding of its hydroclimatic and hydro-meteorological characteristics as well as its hydrodynamics, the same cannot be said for the Vila Nova River. In the case of the Matapi River, for example, the highest net discharge during that particular period took place in April of 2015 [$-434.6 \leq Q \leq 686.5$ m$^3 \cdot$s$^{-1}$], explained by the rainfall incidence of 340 mm (average), the highest of the monsoon season [41]. This had direct implications for the ebb tide flow, which was higher than the flood only in this period, evidencing the influence of rainfall on the runoff and hydrodynamics at the mouth of the Matapi river basin [41].

In the transition period, in June 2015, the flow varied from $-539.5$ m$^3 \cdot$s$^{-1}$ to 436.5 m$^3 \cdot$s$^{-1}$. This represents the rainy-dry transition period, with an average precipitation of 230 mm, much higher than the following months, the only positive difference concerning climatology. In September 2015, it was commented on the influence of the equinox, with a flow range oscillating from $-808.1$ m$^3 \cdot$s$^{-1}$ to 477.0 m$^3 \cdot$s$^{-1}$. Notably, the highest value of total net discharge (1.285.1 m$^3 \cdot$s$^{-1}$) surpassed that of April, despite having only 8 mm of average precipitation at that time. In October 2015, it was estimated that the net discharge was [$-612.8 \leq Q \leq 348.6$ m$^3 \cdot$s$^{-1}$]. Additionally, the average rainfall was 12 mm, slightly higher than in September of that year. It is important to note that these estimations were made without considering the influence of ocean oscillations caused by the equinox [41].

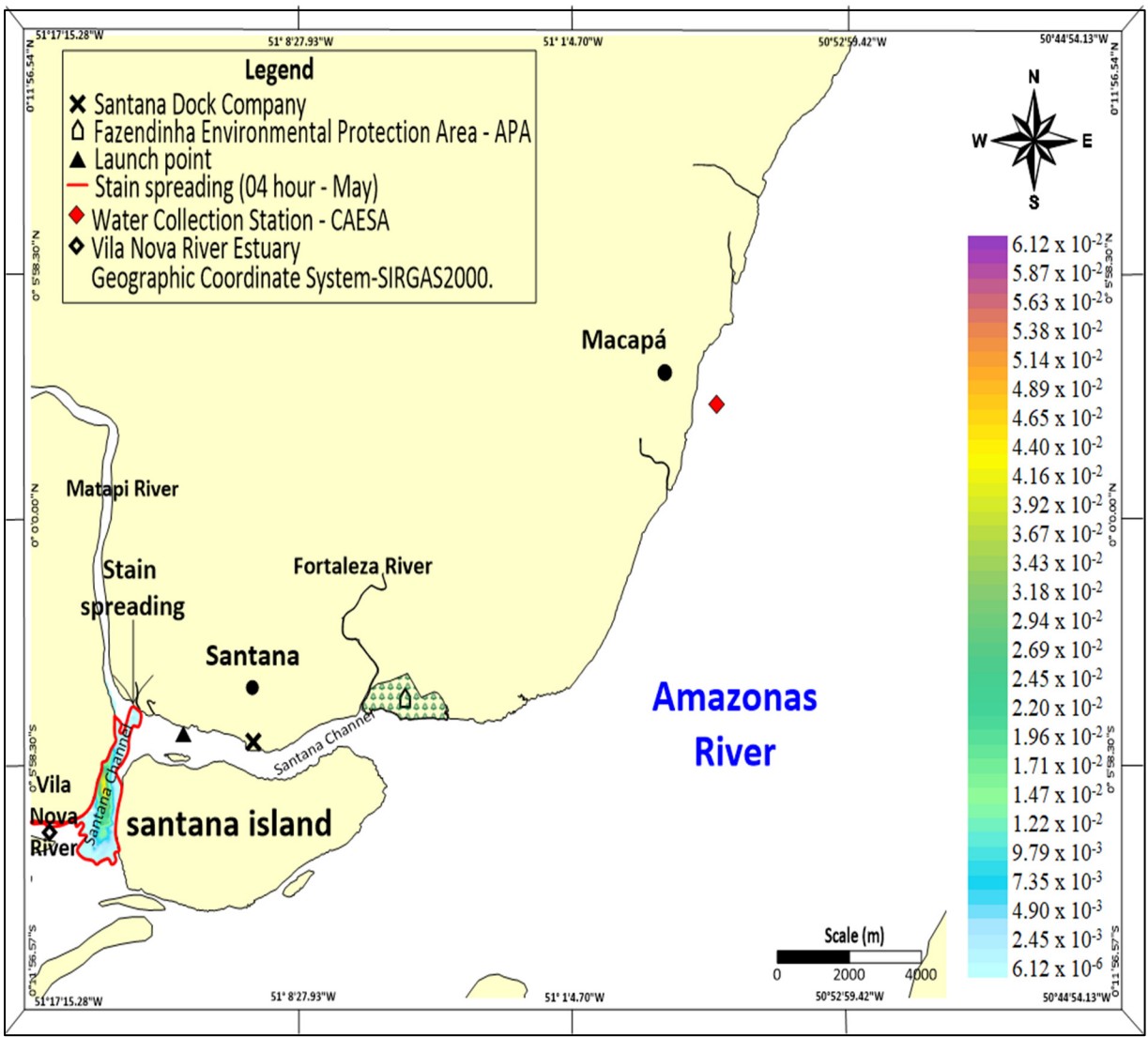

**Figure 6.** S-2b—4 h after the oil spill (rainy season in May). Source: Author, 2022.

In December 2015, at the end of the dry period, the lowest net discharge values observed were recorded [$-568.2 \leq Q \leq 338.3$ m³·s⁻¹]. It can be inferred that the decrease in water levels during this time was a result of the conclusion of the dry season. These values give an idea of the influence of the Matapi River on the Canal de Santana. In any case, there is no data available from flow measurements in the Vila Nova River. However, it is estimated that the latter has a relatively higher flow rate (20 to 40%) than the Matapi River [41].

Returning to the analysis of the present research, in the analysis of the S-2c scenario (Figure 7), it is observed that after 8 h following a potential disaster, with ebb tide characteristics and assuming its highest hydrodynamic potential, the pollutant plume re-enters the Santana channel. Furthermore, due to the peak of the ebb tide, it remained on an eastward trend, almost leaving the channel completely.

The oil has reached its peak concentration and has now spread to the banks of the Fazendinha Environmental Protection Area. In this simulation, the affected area covers 11.29 km², and the recorded flow was 24,013.57 m³/s, which was three times higher than the previous scenario, recording the highest flow in the model. This is important, as it explains the extension traveled by the plume in this analysis interval.

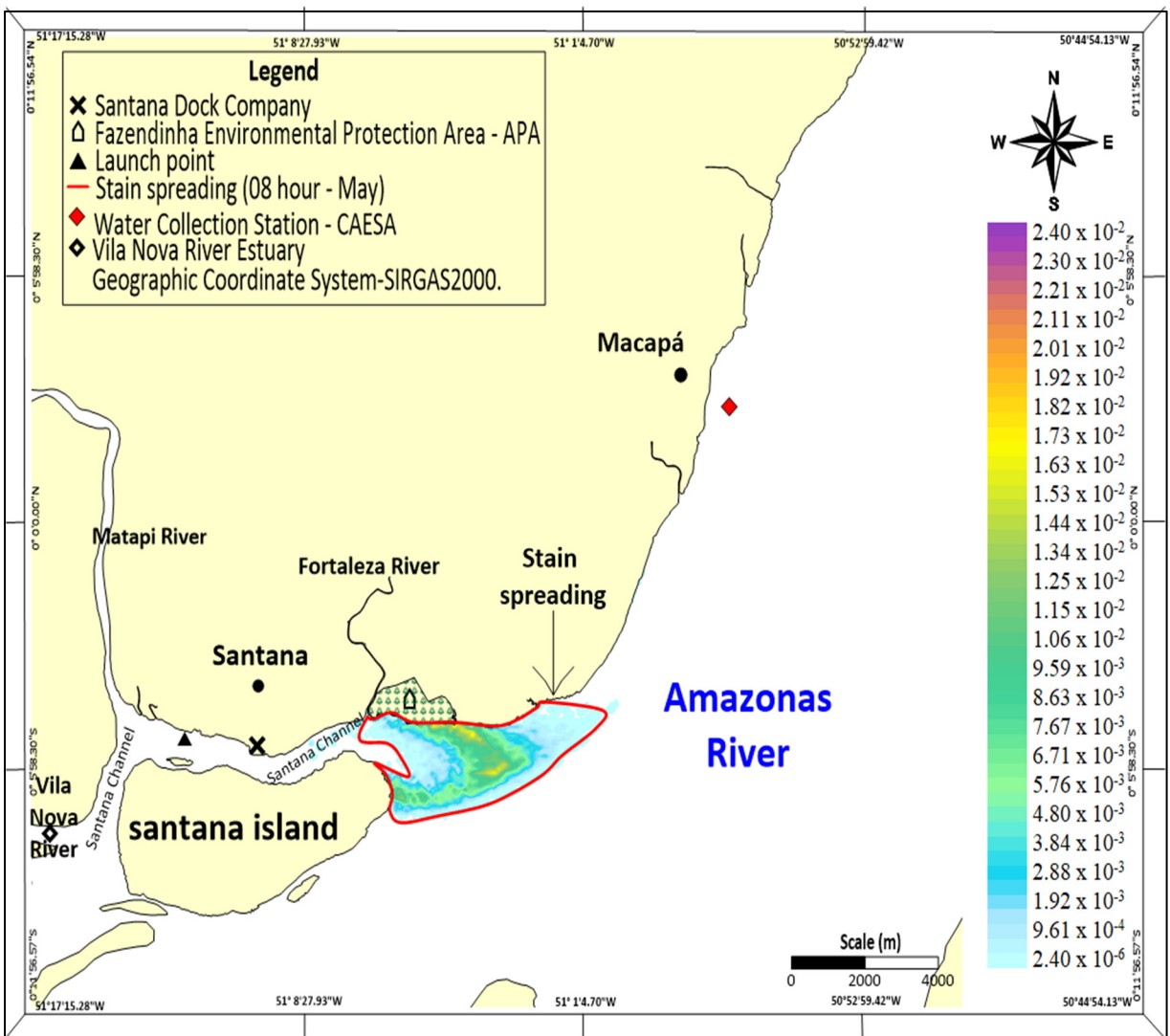

**Figure 7.** S-2c—8 h after the oil spill (rainy season in May). Source: Author, 2022.

In the ebb phases during the rainy season, there is a greater fluvial influence and less influence from marine waters entering the estuary [33,39,40]. Therefore, the recorded flow in this flow direction is higher. The opposite applies in the dry season because the low river level causes the flood tide level to have a greater influence on the amount of water entering the channel [33,40]. This contributes to the inversion of the flow in the Amazonas River [12] and its tributaries [41]. Therefore, this phenomenon was also observed in both scenarios S-1 (a–b) and S-2 (a–b).

The last analysis, at noon, refers to the S-2d scenario (Figure 8), representing the end of the ebb tide phase. In this phase, the plume was concentrated north of the Amazon River, predominantly near the edge of the city of Macapá. In this interval, with a flow of 14,263.46 m$^3$/s, the plume occupied an area of 36.23 km$^2$, recording the worst possible spreading scenario in comparison to the same analyzed times in May. Furthermore, the plume would have already moved completely beyond the urban center of Macapá, concentrating on the northern side of the city, after the Jandiá canal. Therefore, this was considered the worst possible spreading scenario among all analyzed intervals in both seasonal hydrological periods.

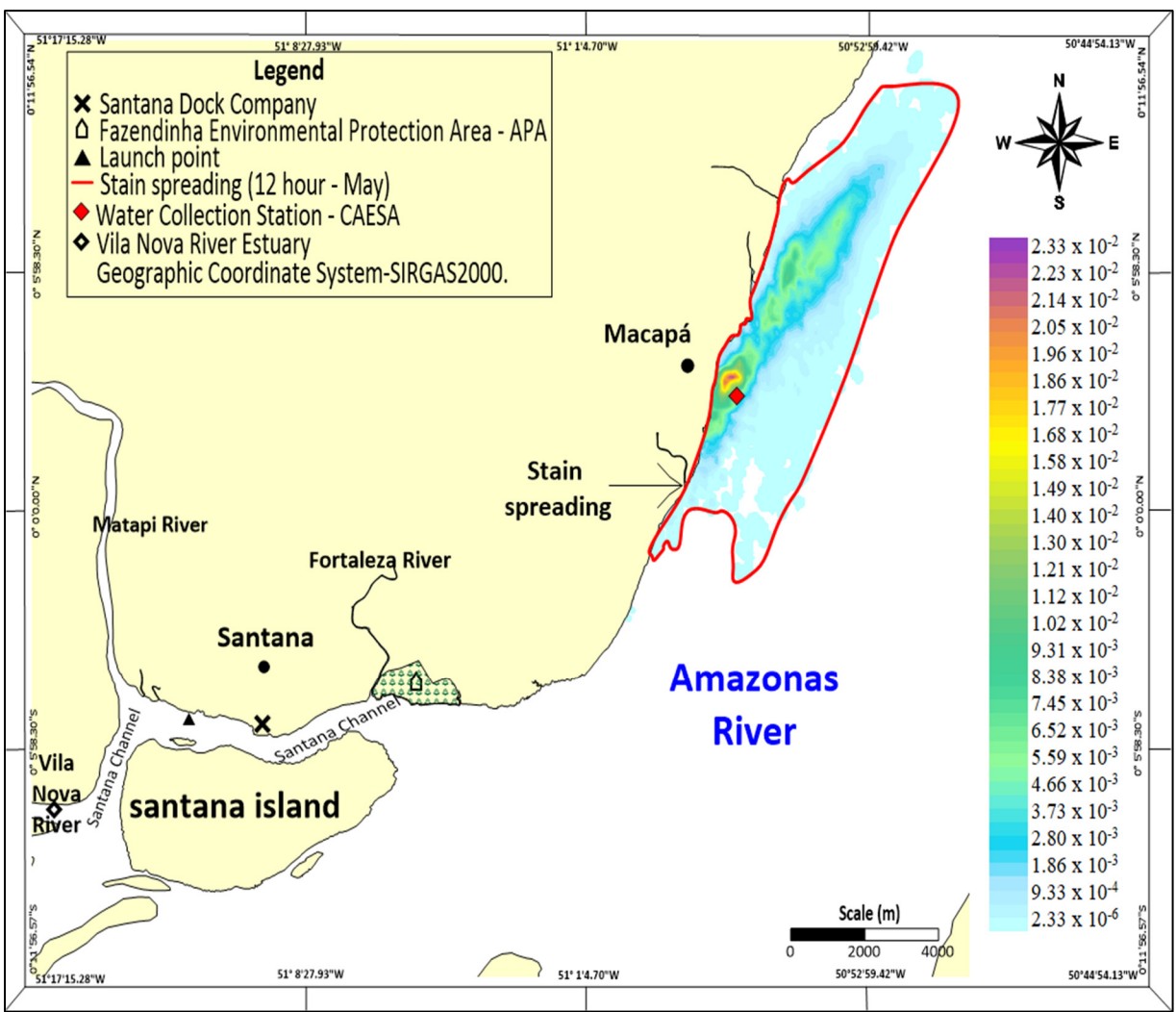

**Figure 8.** S-2d—12 h after the oil spill (rainy season in May). Source: Author, 2022.

In the November scenario (Figures 3 and 4), the plume touched the entire margin of the Fazendinha Environmental Protection Area, while in May the plume extended further downstream, towards the mouth of the Amazon River. Its extension reached the vicinity of the Macapá water intake station facilities, increasing the risks and dangers of environmental and economic impacts and compromising water intake and treatment operations for significant periods [35]. In extreme cases, the Water Treatment Plant—ETA of Macapá would need time and advanced technologies to make the potentially affected water resources less impacted and to make them accessible and useful for consumption again.

Comparing the affected areas in the proposed scenarios with the SAO [42], referring to the studied area, the Littoral sensitivity indices recorded are practically all equal to or higher than 9 (ISL ≥ 9), often reaching ISL = 10 in long stretches. These indicate zones of high and very high vulnerability, encompassing mainly fluvial environments [43]. Thus, only these results along the coastal–estuarine zone would be enough criteria to justify similar analyses to the present study. Even though the limitations of hydrological, meteorological, bathymetric, and water quality experimental data in this region are recognized, they feed the simulation models more reliably [9]. However, the present analysis is one more scientific contribution towards the strengthening of support tools for planning, accident prevention, and management of water resources and coastal ecosystems in the Amazon River estuary in Amapá State [11].

These analyses filled important gaps in knowledge about the hypothetical behavior of a plume of pollutants in the port area of Santana. In the case of accidents, particularly

in situations with very short-term or immediate deadlines, it becomes possible to project the maximum extent of the impacts within these critical time intervals. This highlights the importance of urgent implementation of prevention, mitigation, and environmental control measures. Collaboration among sectors such as water resources management, ecosystem conservation, basic sanitation, and environmental risk analysis becomes crucial in addressing these challenges, particularly in the port area.

We believe that the scenarios generated in this research are extremely useful in the event of a similar local and unannounced disaster (flowing clay). Due to what happens with the simulated hypothetical plumes during only one tidal cycle, it was found that there would be a significant demand for urgent mitigation measures and the control necessary to contain the resulting environmental impacts [9,11,13,43].

In this aspect, it is important to highlight the role of meteorological characteristics that divided the analyses (dry season versus rainy season). However, in these scenarios, the hydro-meteorological criteria were not relevant factors to define the worst situation of contaminant spreading. For example, in the first 4 h, the spreading was wider in the dry period (November) in both scenarios analyzed. On the other hand, after 8 h of spilling, the scenarios registered the worst situation (greater spreading and concentration) in the rainiest period (May). However, it was possible to see from the correlation between flow and plume area that the plume size was proportional to the flow of the respective analysis, as would be expected.

The flow rate was predominant for plume scattering even in the face of the relevance of wind influence in both simulations during the dry period when the latter was compared to previous results [9,12]. This showed some neutrality in the wind effects in May and was not favorable in 25% of the November scenarios. This more favorable wind condition in November [8] is associated with the approximately 25% higher flow volume in the first 4 h. This explains why the plumes in November S-1 (a–b) had the worst pollutant concentration and spreading results when compared to the May S-2 (a–b) scenario.

However, in November, the wind speed is typically more intense. Thus, the wind was associated with the first hours of the spill and had a direct influence on the plume thickness [9,10,23]. This shows the random importance of the instant release of the pollutant into the water (tidal phase), even recognizing that the effect of wind shear stress, the main driver of surface circulation, is gradually lost along the water column [23]. This may have been the main factor culminating in the difference in plume dispersion in the first 4 h of the study between November and May. This difference was about 6 km and 3 km, respectively. It is also important to stress that, precisely in this interval, operations to control and mitigate the impacts of plume expansion also need to be more effective.

In contrast, in scenarios S-1 (c–d) and S-2 (c–d), it was observed that the flow rate in the latter scenario was approximately 30% higher than in the former. This can be attributed to the higher volume of water flowing through the Santana Channel, which is primarily influenced by its river flow. As a result, scenarios S-2 (c–d) configured the highest dispersion capacity of the contaminant plumes in this defined time interval. The 8 h post-spill stage recorded the peak model flow, which made the particles travel approximately 25 km, only during the last 8 h of the simulation. However, in scenario S-1 (a–b), the same interval delimitations registered a travel of only 12 km, ratifying the role of flow in particle transport.

The scope of this possible accident shows the fragility and environmental sensitivity of the area since the impact of an accident would affect extremely vulnerable ecosystems and urban and rural communities [4,36]. The close contact of the contaminant with the shores shown in the simulations is a serious warning about how much the coastal biodiversity would be affected [11,13,29,34]. This area has one of the largest continuous areas of mangroves in the world, increasing the environmental risk of contamination whose impacts would be inestimable from an ecological and socioeconomic point of view [12,14,44].

The eventual exposure of the population to the contaminant was evidenced in this research, demonstrating that ecological, environmental, and socioeconomic vulnerabilities

are evident. Considering that only 12 h after the spill, the water supply of Macapá would already be compromised despite the distance from the accident site, causing problems in the treatment system [9,11,13,35]. In addition, a disaster of medium or large proportions could also collapse a large part of the water supply system and even make local tourism unviable. It causes systemic paralysis of economic activities, such as fishing, leisure, and services related to sanitation and public health, and paralyzes the functioning of the port [4].

The exact timing of the spill is unpredictable when defining the first mitigation actions. The analysis of the eight hypothetical numerical scenarios revealed that the direction of flow plays a significant role in intensifying or reducing the impact of plumes during the spreading process. The intensity of the impact varies according to the flow at each subsequent interval of the tidal phase. Therefore, it becomes a precursor and controller of the dispersive intensity, successively feeding back on itself. This can be observed in both scenarios, especially in the first 4 h after the spill, moments considered essential for any type of mitigating action. However, the following phases can become even more critical. Additionally, without proper control, a wide trail of contamination is formed, reaching wider geographic zones and, consequently, the ecosystems and biodiversity contained in these areas.

## 4. Conclusions

The studies showed that during the dry period (November), the greatest risks were attributed to the flood tide period (S-1a and S-1b). During this time, the tidal flow is opposed to the upstream fluvial flow of the Amazon River and nearby tributaries (the Matapi and Vila Nova Rivers), causing the so-called reverse flow. Once, during the rainy season (May), the typically intense fluvial flow is amplified thanks to the volume of rainwater, intensifying the flow of the Amazon River in its natural direction. In this period, the risks of plume spreading increase due to the ebb tide (S-2a and S-2b).

The hypothesis that plumes, in the very short term of a tidal cycle, significantly affect areas of oil-sensitive slopes classified with [ISL $\geq$ 9.0], thereby elevating the potential for environmental impact, has been confirmed. The geographical setting makes it possible to delineate the affected area using containment barriers. Therefore, if these measures are applied promptly, the chances of mitigation are enhanced.

Considering the 2013 landslide, this study contributes as a subsidy to the immediate risk management of port facilities and energy reserves (fuel tanks) established in the area of the Port of Santana to subsidize initiatives to act in similar environmental accidents in the region, focused on immediate responses. Additionally, it can still have its methodology applied in several important environmental areas, such as the dispersion of contaminants caused by dam breakages.

However, despite the advances of this type of study in the region, there are many limitations to the research, requiring considerable technical, economic, and material investments to undertake new fieldwork and carry out studies in other seasonal hydrological periods. The sustainable development of a region so rich and promising in terms of natural resources and knowledge needs to take every precaution to encourage prior qualitative and quantitative monitoring of oil spills.

The conclusive findings of the study correspond exclusively to the studied scenarios, taking into account their respective hydrodynamic characteristics. The calibrated model used in the study incorporates the effects of oceanic hydrodynamic effects into the initial and boundary conditions of the downstream stretch of the Amazon River. These considerations were implemented consistently throughout the modeling phases and during the simulations of seasonal and tidal variations. Therefore, it is recommended to expand the scope of the research to include other seasonal periods. If possible, an extension and improvement of the study grid should be considered to conduct a more comprehensive and robust analysis with a broader temporal range.

Furthermore, it is recommended to encourage field and experimental work to produce more robust computational meshes. This will help enhance the understanding of local

hydrodynamics and enable a more accurate representation of the flow in the area. By improving the representation of local flow patterns, it will be possible to predict possible environmental accidents more efficiently.

**Author Contributions:** Conceptualization, S.C.D., M.R.T., C.H.M.d.A. and A.C.d.C.; Methodology, S.C.D., M.R.T., C.H.M.d.A. and A.C.d.C.; Software, S.C.D. and C.H.M.d.A.; Validation, S.C.D. and C.H.M.d.A.; Formal analysis, S.C.D., M.R.T., C.H.M.d.A. and A.C.d.C.; Investigation, S.C.D., M.R.T., C.H.M.d.A. and A.C.d.C.; Resources, M.R.T.; Data curation, S.C.D., C.H.M.d.A. and A.C.d.C.; Writing—original draft, S.C.D.; Writing—review & editing, S.C.D., M.R.T., C.H.M.d.A. and A.C.d.C.; Visualization, A.C.d.C.; Supervision, C.H.M.d.A. and A.C.d.C.; Project administration, S.C.D. and M.R.T.; Funding acquisition, S.C.D. and M.R.T. All authors have read and agreed to the published version of the manuscript.

**Funding:** APC support and funding: PROPESP/UFPA (PAPQ). Notice 02/2023. Conselho Nacional de Pesquisa—CNPQ, Process No. 314830/2021-9.

**Data Availability Statement:** Part of the data can be made available, if requested to the corresponding author.

**Acknowledgments:** Conselho Nacional de Pesquisa—CNPQ, Process No. 314830/2021-9, and the support of the Laboratories of Hydraulics and Sanitation of the course of Civil Engineering/UNIFAP and Chemistry, Sanitation, and Modeling of Environmental Systems (LQSMSA) of the course of Environmental Sciences at UNIFAP. Support from PROPESP/UFPA (PAPQ).

**Conflicts of Interest:** The authors declare no conflict of interest.

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
