# Peer review of "Numerical Simulation of Oil Spills in the Lower Amazonas River"

_water, doi:10.3390/w15122197_

Round 1

Reviewer 1 Report

I believe that this manuscript is not well presented and with the introduced form its results can not be generalized. Therefore its novelty and contribution are not sufficient to be published as a technical research paper. Other comments are listed within the attached manuscript file.

Author Response

"Please refer to the attachment."

Good morning,

The esteemed reviewer,  we have attached explanations and reasons, as well as excerpts of suggestions added to the work.

Thank you in advance.

Reviewer 2 Report

Title: NUMERICAL SIMULATION OF OIL SPILLS IN THE LOWER AMAZONAS RIVER

Comment: This study aims to evaluate the potential environmental impacts of an eventual oil spill in the very short term, using a numerical hydrodynamic simulation model coupled with that of pollutant dispersion. This is a very meaningful study.I have the following questions: 1. This study is the data obtained by computer simulation. So, does the calculation model adopted in this study have sufficient basis? What is the comparison with other models? 2. How to verify the simulated data in this study to explain the reliability of the model?

3. The diffusion of oil has a great relationship with the flow of ocean. Does the author take into account seasonal changes in ocean currents to assess oil diffusion?

Author Response

(The authors gave the same response as above.)

Reviewer 3 Report

The manuscript is based on a meny of assumptions. Some factors were assumed without a precise justification for the choice. I have the impression that the simulations used do not reflect the real conditions. I don't think the manuscript will be of interest to readers from distant lands. However, the presented results may be of local significance. I therefore propose to publish the presented research in a local scientific journal.

Author Response

"Please refer to the attachment."

Round 2

Reviewer 1 Report

The manuscript still needs to be improved considering the following  comments:

1. What is the main question addressed by the research?
2. What is the topic original or relevant in the field? What is
the specific gap in the field that has been covered?
3. What does it add to the subject area compared with other published material?
4. What is new and improvement about the methodology? Can it be generalized?
5. Are the conclusions presented in a generalized form? The answer is no where it must be generalized.
6- do they address the main question posed?
All these comments need to be addressed in the manuscript which is a difficult task but I will give major review to the authors as an option that may the editor may consider in his dicision.

Author Response

Dear, The contributions and English translation and proofreading certificate follow. Yours sincerely, Author.

Reviewer 2 Report

Although the quality of the article can still be improved, from the author's response, the author has seriously revised and replied. I think the article can be published.

Author Response

Dear, Contributions are answered below. Yours sincerely, Author.

Reviewer 3 Report

The authors provided exhaustive answers. The manuscript needs to be checked against MDPI editorial requirements. After minor corrections, the manuscript can be accepted.

Author Response

(The authors gave the same response as above.)
